# Cell Cycle-Dependent Control and Roles of DNA Topoisomerase II

**DOI:** 10.3390/genes10110859

**Published:** 2019-10-30

**Authors:** Joyce H. Lee, James M. Berger

**Affiliations:** Department of Biophysics and Biophysical Chemistry, Johns Hopkins University School of Medicine, Baltimore, MD 21205, USA; jlee529@jhmi.edu

**Keywords:** topoisomerase II, cell cycle, mitosis, DNA replication, decatenation, cell cycle checkpoint, cancer

## Abstract

Type II topoisomerases are ubiquitous enzymes in all branches of life that can alter DNA superhelicity and unlink double-stranded DNA segments during processes such as replication and transcription. In cells, type II topoisomerases are particularly useful for their ability to disentangle newly-replicated sister chromosomes. Growing lines of evidence indicate that eukaryotic topoisomerase II (topo II) activity is monitored and regulated throughout the cell cycle. Here, we discuss the various roles of topo II throughout the cell cycle, as well as mechanisms that have been found to govern and/or respond to topo II function and dysfunction. Knowledge of how topo II activity is controlled during cell cycle progression is important for understanding how its misregulation can contribute to genetic instability and how modulatory pathways may be exploited to advance chemotherapeutic development.

## 1. Introduction

Cell proliferation requires that the genetic information from a starting mother cell be copied and partitioned into two daughter cells. The double-stranded nature of DNA enables semiconservative replication [1]; however, unwinding of the DNA helix produces superhelical tension that poses a topological challenge to replicative processes [2] and leads to entanglement of newly-replicated sister chromosomes [3]. In all branches of life, enzymes known as topoisomerases help cells circumvent these and other topological challenges. Topoisomerases broadly fall into two principal categories, termed type I or type II, based on whether they respectively form single- or double-strand DNA breaks during catalysis [4]. Although both type I and type II topoisomerases can alter DNA superhelicity [5,6,7], type II enzymes are uniquely able to decatenate intact sister chromosomes by an ability to pass one double-stranded segment of DNA through a transient, enzyme-mediated break in a second DNA duplex [8,9].

Most eukaryotic organisms express a single type II topoisomerase known as DNA topoisomerase II (topo II or Top2). Vertebrates carry two isoforms, topo IIα and topo IIβ [10,11]. Topo IIα is primarily active during DNA replication and mitosis [12], whereas topo IIβ has roles in transcriptional regulation of gene expression [13,14,15]. Topo II (or topo IIα) activity during mitosis is essential for cell viability [16]. Upon entry into S-phase, topo II is critical for organizing genome structure, promoting chromosome segregation, and preventing aberrant entry into anaphase with partially decatenated sister chromatids. Topo II activity is modulated by various means in response to the needs of the cell. For example, the abundance of topo IIα in vertebrates is cell cycle-regulated to accommodate the substantial changes in chromosome copy number and structure that occur during DNA replication and mitosis [17,18,19]. The affinity of topo II for DNA substrates and catalytic activity can also be directly controlled through cell cycle-dependent changes in DNA topology, protein–protein interactions, and post-translational modifications [20,21,22]. Missteps in topo II-dependent processes activate a decatenation checkpoint that stalls the cell cycle at the G2-to-M transition and is protective against genomic damage [23]; loss of this checkpoint has been linked to tumorigenesis [24]. The present review seeks to provide a brief overview of the various roles of topo II and of known cell-cycle regulatory mechanisms that help monitor and coordinate topo II activity with the dynamic genomic landscape. Due to limitations on space, recent developments concerning the action and consequences of topo IIβ in supporting gene expression will not be discussed.

## 2. Cell Cycle-Dependent Expression of Topo IIα in Vertebrates

In vertebrates, production and degradation of topo IIα is cell cycle-regulated and serves to link topo II functions with proliferation. Topo IIα levels increase in mid-S phase through mitosis and rapidly decrease upon mitotic completion (Figure 1) [18,19,25]. Evidence suggests that accumulation of topo IIα is due to upregulation of *TOP2A* transcription and mRNA stabilization. For example, the promoter region of *TOP2A* contains multiple inverted CCAAT boxes (ICBs) that have been proposed to interact with the NF-Y, SP1, and ICBP90 transcription factors to regulate the cell cycle-dependent expression of topo IIα [26,27,28,29]. Stabilization of *TOP2A* mRNA by the 3′UTR from S phase through mitosis has also been observed in human cells, though not in mouse cells. Studies have suggested that the 3′UTR of human *TOP2A* mRNA binds to cell cycle-regulated proteins to protect it from degradation [30,31]. At the protein level, topo IIα abundance is controlled by ubiquitin-mediated proteasomal degradation [32]. Depletion of ubiquitin specific protease 15 (USP15), a deubiquitylase (DUB), has been associated with a failure to appropriately accumulate topo IIα, while USP15 expression correlates with topo IIα expression [33]. These observations suggest that USP15 stabilizes topo IIα by removing ubiquitin modifications. After mitosis, topo IIα appears to be ubiquitylated by the anaphase promoting complex (APC) and Cdh-1 in the absence of USP15, resulting in a rapid decrease in enzyme concentration [34]. Cell cycle-dependent expression of topo II has not been observed in *Drosophila*, which carry only one isoform [35]. In vertebrates, topo IIβ is present at uniform levels throughout the cell cycle [19]. Expression of topo II in organisms that carry a single isoform may not be proliferation-dependent because a single enzyme must assume the roles of both topo IIα and topo IIβ.

## 3. DNA Replication

During replication, origin unwinding and fork progression generate positive writhe ahead of the replisome. The accumulation of superhelical tension, if not relieved, can lead to fork stalling, the formation of precatenanes, and DNA damage [2]. In eukaryotes, supercoil relaxation by topoisomerase I (topo IA) is generally sufficient to support replication, as the loss of topo II does not appear to appreciably lower rates of DNA synthesis [36,37]. However, in *S. cerevisiae* at least, topo II is able to compensate for the loss of topo I, indicating that topo II is active at these early stages of DNA replication [36]. The simultaneous deletion of topo I and topo II causes genomic instability [36]. To prevent DNA damage from the accrual of superhelical tension, both topo I and topo II are rapidly recruited to sites of origin firing during replication initiation [36]. While topo II is not required for DNA replication initiation in a G1-to-S phase transition, inhibition of topo II has been observed to prevent entry into S phase from quiescence (G0) in mammalian cells and *Drosophila,* but not in yeast [38,39]. After initiation, topo II assists with supercoil relaxation and the disentanglement of newly replicated segments of DNA during replication elongation, as most decatenation is complete by entry into prophase (Figure 2A) [40,41]. 

The role of topo II in replication termination is less well-understood. ChIP studies in *S. cerevisiae* have indicated that topo II preassociates with common replication termination sites [42]. However, unlike prokaryotic systems [43,44], topo II activity is not required for fork convergence or the dissolution of replication forks [45,46]. Topo I and topo II alone are insufficient for efficient termination; instead, recent work has suggested that replication termination may primarily rely on Pif1-family helicases [46]. Topo II activity may be more critical in telomeric regions that are prone to replication stalling due to highly repetitive sequences [47]. Stalled replication forks generate telomere-specific extrachromosomal structures that may require topo II to facilitate fork progression and replication termination [48,49]. This observation suggests that topo II may expedite termination in regions prone to replication stress, but that it is not generally essential for replication termination.

In eukaryotes, DNA replication is initiated asynchronously from multiple origins and some origins remain dormant throughout S-phase [50,51]. To prevent re-entry into S-phase, cells must dismantle replication machineries. Interestingly, in *Xenopus*, depletion of topo IIα has been observed to prevent the complete degradation of origin recognition complex subunit 1 (ORC1) and to block the dissociation of ORC1/2 from replication protein A (RPA) following the completion of DNA synthesis [52]. Thus, topo II may participate in preventing replisome reassembly at the end of S phase. Overall, topo II plays a nonessential role in DNA strand synthesis but seems to nonetheless facilitate this process in problematic regions of the genome. 

## 4. Chromosome Condensation and Sister Chromatid Cohesion 

Early studies by Earnshaw et al. [53] and Gasser et al. [54] were the first to identify topo II as a component of the mitotic scaffold and suggest a structural role for topo II in mitotic chromosome organization. Subsequent studies have shown that topo II is required for chromosome condensation but not for maintenance of condensed chromosomes [55]; i.e., the activity of topo II appears to facilitate chromosome compaction but the protein itself does not seem to stably link genomic regions together. The inhibition or depletion of topo II activity partially rescues chromosome compaction defects when cohesin (a structural maintenance of chromosomes (SMC)-family complex) is disrupted, indicating that decatenation and supercoil relaxation by topo II can antagonize condensation [56,57]. On the other hand, loss of topo II function also leads to incomplete chromosome condensation, suggesting an active role for topo II in this process [17,58,59,60]. Together, these findings indicate that a cell must delicately balance topo II activity to appropriately compact chromatin in early mitosis. This ‘Goldilocks’ equilibrium is maintained by the interplay between condensin and topo II (Figure 2B) [61]. Condensin is required for chromosome condensation and can generate positive writhe in an ATP-dependent manner [62,63]. The induction of positive supercoils by condensin has been suggested to contribute to chromosome compaction in prophase. Conversely, topo II rapidly relaxes positive supercoils. Thus, the cell may tune topo II and condensin activity to maintain appropriate chromosome condensation. 

Regulating topo II decatenation activity is also critical for sister chromatid cohesion. Sister chromosomes are generally kept in close proximity to one another, which can aid homologous recombination in repairing DNA damage with minimal error [64]. Although decatenation is generally favored by type IIA topoisomerases [8], topo II can introduce catenanes as well (Figure 2B) [65,66]. In cells lacking condensin I, topo II serves to recatenate regions of sister chromosomes that are in close proximity [67]; overexpression of topo II further increases sister chromatid intertwining [68]. Most genomic regions are decatenated by topo II before mitotic entry; however, specific chromosomal regions such as centromeres, telomeres, and rDNA regions remain in a partially catenated state until anaphase [69,70,71,72]. Several studies have observed increased binding of topo II at centromeres after chromosome condensation [73,74,75]. Despite this enrichment, centromeric DNA is protected from decatenation by cohesin, which stays bound until the onset of chromosome segregation (Figure 2C) [57,69,76]. Topo II has also been suggested to play a structural role in cohesin loading [77]. Once cohesin is cleaved by the enzyme separase, centromeric DNA becomes exposed to topo II (Figure 2D) [78]. Prior recruitment of topo II to centromeres enables the timely resolution of these regions so that chromosome segregation can occur without the formation of interchromosomal DNA linkages known as anaphase bridges [57,69,76]. In this manner, sister chromatid cohesion is partially controlled by regulating topo II decatenation activity. 

## 5. Chromosome Segregation

Interlinked sister chromosomes are fully resolved from one another in anaphase, as the mitotic spindle pulls each chromosome to opposing poles of the cell (Figure 3, upper) [79]. However, volumetric analysis of overlap between sister chromosomes has indicated that most of this resolution occurs simultaneously with chromosome condensation in late prophase and then continues through the cell cycle [80]. This observation is in agreement with studies showing that a majority of precatenanes are removed during DNA replication prior to mitotic entry [40,41]. Catenanes that persist until anaphase are generally found in regions with highly repetitive sequences and are hypothesized to contribute to sister chromosome cohesion [69,70,71,72]. Chromosome length also appears to correlate with catenane retention into late anaphase [81]. Failure to untangle these regions leads to the formation of anaphase bridges (Figure 3, middle) [82,83,84,85,86]. If left unresolved, anaphase bridges can promote genomic damage [87]. 

Topo II and condensin are recruited to centromeric regions as cells progress through mitosis so as to prime these sites for timely decatenation [73,85]. Similar to a loss of topo II activity, disruption of the condensin complex leads to the formation of anaphase bridges [17,41,88,89,90]. The loss of condensin decreases topo II enrichment at centromeres, suggesting that condensin aids in recruiting topo II to these regions [85]. How condensin and topo II interact is still debated, as some studies from yeast and *Drosophila* suggest a physical interaction [84,89], while others have observed that the two proteins do not colocalize on condensed chromosomes in *Drosophila* [91]. Regardless, studies have shown that condensin stimulates topo II activity both in vitro and in vivo [84,89]. Independent of direct binding to topo II, condensin can enhance the decatenation activity of topo II by increasing positive writhe to present a more favorable DNA substrate for decatenation as opposed to supercoil relaxation [20,92]. ATP-dependent formation of positive supercoils by condensin is cell cycle regulated by mitosis-specific phosphorylation events executed by Cdc2-cyclin B, as well as by a requirement for mitotic spindle attachment [20,93]. The cell cycle-dependent accumulation of positive writhe has been observed in budding yeast depleted of topo II as catenated plasmids transition from a negatively-supercoiled state (in the pre-mitotic stage) to a positively-supercoiled state (in mitosis) [20]. This topological transition towards positive writhe does not occur under nocodazole treatment, suggesting that mitotic spindle attachment may also regulate this process [20]. Human topo IIα has been additionally shown to preferentially relax positive supercoils, whereas topo IIβ has no bias [94]. Thus, the accumulation of positive writhe in a cell cycle-dependent manner through condensin activity increasingly promotes decatenation in centromeric regions as mitosis progresses.

Like centromeric regions, rDNA regions and telomeres frequently sustain partial catenation until anaphase onset. In addition to topo IIα and condensin, the resolution of rDNA regions in mammals involves tankyrase I, the Plk1-interacting checkpoint helicase (PICH), and topoisomerase 3α (topo IIIα) [70,71]. Tankyrase I is a polyADP-ribose transferase that is thought to interact with condensin II to stimulate topo II decatenation activity [71]. PICH is a SNF2 family DNA translocase that localizes to anaphase bridges formed in rDNA regions [70,83]. Topo IIIα, a type I topoisomerase, appears to cooperate with PICH to generate positive writhe in an ATP-dependent manner similar to condensin [86]. Disrupting PICH/topo IIIα positive supercoiling activity has been shown to generate anaphase bridges in rDNA regions. In addition to PICH, topo III interacts with RecQ family proteins in eukaryotes to facilitate DNA damage repair [95]. In yeast, *Drosophila*, and mammalian cells, the loss of different types of RecQ proteins can cause mitotic defects and the formation of anaphase bridges [96,97,98,99]. Potential physical interactions between RecQ helicases and topo II have been detected in yeast and humans, and a correlation between RECQ5L levels and topo IIα decatenation activity has been reported as well [98,99]. Future studies will help reveal the mechanistic details of the interactions between topo II, topo III, and RecQ and the roles of these interactions in facilitating chromosome segregation. With regards to telomeric regions, components of the shelterin complex, TRF1 and TRF2, have been proposed to affect topo II localization to telomeres [49,72]. Additional work is needed to uncover the mechanisms that control decatenation status in rDNA and telomere regions.

## 6. Post-Translational Modifications

Post-translational modifications on topo II can modulate protein stability, catalytic activity, and binding affinity [4]. SUMOylation, in particular, has been described as a key modification for regulating topo II in a cell cycle-dependent manner. In budding yeast, SUMOylated topo II has been found to be highly enriched in cells arrested at metaphase, suggesting that this modification is cell cycle-dependent [100]. Although yeast only harbor a single SUMO gene, vertebrates have multiple genes that encode different isoforms. Several studies have reported that topo II from *Xenopus* is primarily modified by SUMO-2/3 [101,102,103,104]. SUMO-2/3 modifications accumulate upon mitotic entry and are rapidly deconjugated in anaphase [104]. Disruption of SUMOylation prevents the proper localization of yeast topo II and murine topo IIα to centromeres after chromosome condensation [105,106]. Overall, defects in the SUMO-dependent modification of topo II appear to lead to errors in chromosome condensation, sister chromatid cohesion, and chromosome segregation [100,103,104]. 

The exact pathways responsible for SUMOylation of topo II are still debated. Several studies have provided evidence of topo II modification by different SUMO E3 ligases, albeit with sometimes conflicting results. PIASγ-dependent SUMOylation of *Xenopus* topo IIα on Lys660 has been observed in vitro [101]. In *Xenopus*, depletion of PIASγ has been reported to cause a loss of SUMO-2/3 modified topo IIα and lead to severe chromosome segregation defects in anaphase [103]. Inhibition of topo II SUMOylation in the C-terminal domain (CTD) by SIZ1/SIZ2, yeast homologs of PIAS, has been observed to impede proper segregation of centromere-containing minichromosomes [105]. However, it has also been reported that the loss of PIASγ in murine cells has no effect on topo IIα SUMOylation status and does not lead to any mitotic defects [106]; in this instance, RanBP2, a nucleoporin with SUMO E3 ligase activity, appears to be the primary E3 responsible for SUMOylation of topo II [106]. In murine cells, depletion of RanBP2 has been noted to lead to severe aneuploidy, the formation of anaphase bridges, and loss of topo IIα SUMOylation. Unmodified topo IIα also did not appear to properly localize to centromeres of condensed chromosomes in this study, suggesting that SUMOylation may regulate the interaction of topo IIα with chromatin during mitosis [106]. In the case of both PIASγ and RanBP2, the SUMOylation of topo II seems to control enzyme function during chromosome segregation. 

More recent work has indicated that human topo IIα is also SUMOylated at Lys1520 by the non-SMC element 2 (NSE2) subunit of the SMC5/6 complex [107,108]. ICRF-193, a catalytic inhibitor of topo II, induces SUMOylation of topo II and cell-cycle arrest in G2 [23,108], but in NSE2 mutant cells, G2 arrest was disabled and ICRF-193 dependent SUMOylation of topo IIα was reduced, suggesting that SUMOylation is a response to topo II inhibition [108]. Less is known about the deconjugation of SUMO modifications from topo II. Deletion of Smt4, an isopeptidase responsible for SUMO deconjugation in yeast, has been associated with defective cohesion of centromeric regions [100]. This phenotype may be related to retention of SUMO on topo II, as removal of SUMOylation sites on topo II partially rescues the cohesion deficiency [100]. Based on our current understanding, it is likely that topo II is targeted by multiple SUMO E3 ligases at different sites and/or times in its action. 

SUMOylation can affect topo II function in a variety of ways, the effects of which differ depending on the site of the modification. For example, SUMOylation in the C-terminal region of yeast topo II or human topo IIα has been shown to increase affinity for centromeric regions [105,109]. By contrast, SUMOylation of *Xenopus* topo IIα has been observed to decrease affinity for chromatin in general [104]. Together, these observations suggest that SUMOylation redistributes topo II on chromatin to enrich binding to centromeric regions, particularly during mitosis. In addition to altering the affinity of topo II for chromatin, SUMOylation also enables topo II to recruit other proteins to the centromere. In both yeast and *Xenopus,* the SUMOylated CTD has been shown to associate with Haspin (a histone H3 kinase) [110,111], an interaction that is stabilized by cell cycle-dependent phosphorylation of Haspin [110,111]. These studies hypothesize that upon binding to topo II/topo IIα, Haspin catalyzes the phosphorylation of Thr3 on histone H3 in centromeric nucleosomes to recruit the Aurora B kinase [110,111]. The localization of the Aurora B kinase at centromeres is in turn crucial for kinetochore assembly, mitotic spindle attachment, and monitoring of the spindle assembly checkpoint [112]. Claspin, a protein that can activate checkpoint kinase 1 (Chk-1) in response to errors in DNA replication [113], also binds to the SUMOylated *Xenopus* topo IIα-CTD in centromeric regions and has been proposed to be involved in activating Chk-1 upon delays in decatenation [114]. These SUMO-dependent interactions contribute to the structural role of topo II in organizing centromeric complexes during mitosis. 

Ubiquitylation represents yet another modification that can control not just topo II activity but also its abundance. In mammalian cells, ubiquitylation of topo IIα by Cdh-1 is thought to signal for the proteosomal degradation of the enzyme [34]. Cdh-1 is known to interact with the APC to target several cell cycle regulators for degradation following mitotic exit to prevent re-entry into S-phase [78]. Degradation of topo IIα in G1 likely serves to prevent inappropriate enzyme activity that might cause genome instability prior to initiation of DNA replication. Ubiquitylation of topo IIα also has been suggested to directly alter enzyme activity and chromatin affinity. For example, RNF168 and BRCA1, two proteins associated with DNA damage response, have been proposed to regulate ubiquitylation of topo IIα to promote topo IIα binding to chromatin and DNA decatenation [115,116]. Downregulation of RNF168 leads to defects in chromosome condensation, decreased ubiquitylation of topo IIα, and impaired response to topo II inhibition [115]; USP10 is thought to be a DUB that counteracts RNF168 action on topo IIα [115]. By comparison, BRCA1-mediated ubiquitylation of topo IIα has been proposed to stimulate decatenation activity and mobility on chromatin [116]. At present, the exact sites of ubiquitylation on topo IIα have not yet been identified. There remains much to be discovered to understand how this modification regulates topo II. 

Phosphorylation is a third type of modification that is important for proper localization of topo II, regulation of topo II activity, and response to topo II inhibition. Phosphorylation of topo II has long been known to increase in G2 and M phase [117,118]. Many kinases such as casein kinase 1 (CK1), casein kinase 2 (CK2), protein kinase C (PKC), Ca^2+^/calmodulin dependent kinase II (CaMKII), Cdc7, and polo-like kinase 1 (PLK1) have been proposed to target topo II in vitro and in vivo [118,119,120,121,122,123,124,125]. Phosphorylation of topo II by CK2 has been the best characterized to date, and the majority of its phosphorylation sites identified thus far have been mapped to the CTD of the enzyme [126,127]. Phosphorylation of *Drosophila* topo II in the CTD and mammalian topo IIα in the CTD and N-terminal ATPase domain has been reported to enhance enzyme activity in vitro [121,124,128,129]. However, abolishing phosphorylation of human topo IIα by CK2 has been shown to have no effect on catalytic activity [122]. Because many phosphorylation events occur on the highly divergent CTD of topo II, it is possible that there are species-specific and site-specific effects on enzyme activity. The inhibition of phosphorylation has been shown to exacerbate chromosome missegregation during treatment with the topo II ATPase inhibitor, ICRF-193 [120,121,127]. The loss of phosphorylation sites in murine topo IIα appears to inhibit SUMOylation at a nearby lysine and results in weakening the interaction of topo IIα with centromeric regions [109]. Phosphorylation of human topo IIα has been also suggested to maintain stable G2/M-specific interactions with mediator of DNA damage checkpoint protein 1 (MDC1) and Geminin [125,130]. These interactions may be important for regulating topo IIα mobility on chromatin and/or sensing topo IIα dysfunction [125,130]. Phosphorylation has also been shown to regulate topo II interaction with 14-3-3ε [131]. 14-3-3 family proteins recognize phosphorylated targets and 14-3-3ε has been implicated in regulating the G2/M transition [132]. These data suggest a potential signaling role for phosphorylation upon a loss of topo II activity. Overall, topo II/topo IIα is likely a substrate for many different kinases and generally, phosphorylation events are more frequent in S phase through mitosis as compared to G1. However, there are several phosphorylation sites, many of which are found in the nonconserved C-terminal region. It is clear that phosphorylation serves a function in cell cycle-dependent regulation of topo II, but the pathways involved are highly complex and may vary between species; as such, the physiological impact of this modification remains enigmatic. 

## 7. Decatenation Checkpoint 

Eukaryotic cells possess a “decatenation checkpoint” that acts independently of their DNA damage checkpoint and spindle assembly checkpoint (SAC) [133]. Chemical inhibition of topo II by ICRF-193 or the expression of a catalytically compromised topo II leads to cell cycle arrest at the G2/M transition and the formation of chromosomal aberrations (Figure 3, lower) [23,134,135]. However, the depletion of topo II itself allows cells to bypass the decatenation checkpoint and proceed through mitosis without fully segregating their chromosomes (Figure 3, middle) [136,137,138]. The available data indicate that the decatenation checkpoint senses the state of the topo II protein rather than the decatenation status of DNA. 

The full pathway for activating the decatenation checkpoint has not been determined; however, several key components have been identified. In mammalian cells, both topo IIα and topo IIβ appear to be required for efficient initiation of G2 arrest but play counteracting roles in this respect [139]. Topo IIα and topo IIβ are highly homologous in their ATPase domains and catalytic cores, but relatively divergent in their CTDs [140]. The CTDs of topo IIα and topo IIβ likely contribute to their different roles in monitoring decatenation status [139]. The regulatory functions of the CTDs are in part linked to the many post-translational modifications that occur in this region. Both SUMOylation and the phosphorylation of C-terminal residues of topo IIα are required for activation of the decatenation checkpoint [108,130]. These modifications may regulate interactions with proteins that signal for cell cycle arrest. Components of both the DNA damage checkpoint and the SAC (Rad9a, Mad2, Chk1, ATR, and BRCA1) have been suggested to participate in activating the decatenation checkpoint [56,141,142,143,144]. Chemical inhibition of the p38 mitogen-activated protein kinase (MAPK) has been observed to inactivate the decatenation checkpoint, implying that signaling for the decatenation checkpoint involves the p38 and the MAPK pathway [145]. Mitotic entry is triggered by activation of cyclin B-Cdk1 followed by transport of the cyclin B-Cdk1 complex into the nucleus [146]. Expression of a constitutively-nuclear cyclin B construct has been seen to override the mitotic delay induced by ICRF-193, suggesting that the decatenation checkpoint may operate by sequestering cyclin B/Cdk1 in the cytoplasm [143]. Continued efforts are required to determine the interplay between all the factors, known and to-be-discovered, that contribute to activation of the decatenation checkpoint. 

While there is overlap between the decatenation checkpoint and other cell-cycle checkpoints, the response to incomplete chromosome decatenation is clearly distinct from the SAC and the DNA damage checkpoint. SAC activation appears to be unaltered in topo II temperature-sensitive mutants treated with nocodazole and SAC signaling does not occur at nonpermissive temperatures in the absence of this drug, demonstrating that the SAC is independent of topo II activity [111]. A marker of SAC activation is the recruitment of Mad2 and Bub1 serine/threonine-protein kinase to kinetochores [147]. Although Mad2 is required for the decatenation checkpoint, the retention of Mad2 and Bub1 on kinetochores after ICRF-193 treatment has not been detected [141]. When cells transition from metaphase to anaphase after spindle assembly, the mitotic checkpoint complex dissociates from the kinetochores to trigger ubiquitylation and the subsequent degradation of securin, a negative regulator of the separase protease [147]; after its release from securin, separase is able to cleave cohesin to trigger sister chromatid separation [147]. Cell cycle delay induced either by ICRF-193 or by a loss-of-function topo II mutant has been observed even in cells lacking securin, further indicating that the decatenation checkpoint is separate from the SAC [133,134]. Similarly, several lines of evidence have suggested that the decatenation checkpoint is independent of the DNA damage checkpoint [23,141]. Depletion of topo IIα or the expression of a CTD-less topo IIα disables the decatenation checkpoint but both have been shown to leave the DNA damage checkpoint intact [130,148]. Although ICRF-193 has been suggested to induce DNA damage in some cases [149,150,151], cells with an impaired DNA damage response have been seen to arrest after ICRF-193 treatment, supporting the idea that cell cycle arrest upon ICRF-193 treatment is not a result of drug-induced DNA damage [152]. Although ATR is necessary for ICRF-193-induced G2 arrest, both the ATM and Rad53 checkpoint kinases are not [134,143,145]. Based on these observations, it is clear that the decatenation checkpoint represents a distinct control pathway from other known checkpoints. 

Like all cell-cycle checkpoints, the decatenation checkpoint is protective against genomic damage. Chromosomal instability and aneuploidy can often drive tumorigenesis [153], and these gross chromosomal segregation errors are frequently observed in cells lacking an operational decatenation checkpoint [137,138]. Multiple human cancer cell lines have been described to have defective decatenation checkpoints [152,154,155]. Progenitor cells and stem cells also seem to lack the decatenation checkpoint, and this deficiency has been proposed to contribute to the formation of cancer stem cells that drive tumorigenesis [156]. Topo IIα appears to directly interact with the Brg1-associated factors (BAF) chromosome remodeler complex [157], subunits of which are frequently mutated in malignancies [158]. Interestingly, mutations in the Brg1 subunit of the BAF complex have been shown to lead to the formation of anaphase bridges [157]. Based on these observations, it has been hypothesized that BAF-related cancers may arise from dysregulation of topo IIα. Cancer cells with impaired decatenation checkpoints are often highly sensitized to topo II inhibitors, many of which are already clinically approved [152,155,159]. Furthermore, small molecules that alter the decatenation checkpoint response have been described and may increase the efficiency of existing topo II-targeted chemotherapeutics [160]. Thus, identification of cancers with functioning vs. nonfunctioning decatenation checkpoints may aid in the design of more targeted cancer treatments.

## 8. Discussion 

Topo II is a versatile molecular machine that can be tuned to serve the needs of a changing cellular environment. Based on biochemistry and structural biology studies, many (though certainly not all!) aspects of topo II mechanism are becoming reasonably well understood. However, there is still much to be learned about how topo II function is differentially regulated as the requirements for its activity change throughout the cell cycle. Topo II activity is adjusted and targeted very precisely by regulating protein abundance, binding interactions, changes in DNA substrate topology, and post-translational modifications (Figure 4). Topo II also appears to provide a structural or scaffolding capability that can help recruit other proteins to specific chromosomal regions. Cells carefully monitor topo II, possibly by recognizing post-translational modifications or conformational changes that occur upon binding to DNA, as an indication of decatenation status to prevent anaphase entry prior to the complete resolution of sister chromosomes. Understanding the regulatory pathways involved in the control of and response to topo II activity has direct clinical implications, as flaws in these pathways are frequently detected in human cancers [24,152,154,155]. Current topo II-directed chemotherapeutics are effective but limited in their use due to side effects such as the formation of secondary malignancies and cardiotoxicity [159]. Deeper insights into topo II regulation may reveal genetic markers to identify cancers that are sensitized to topo II inhibitors, as well as novel drug targets and treatment strategies that would help increase the efficacy of currently-available therapies.

## Figures and Tables

**Figure 1 genes-10-00859-f001:**
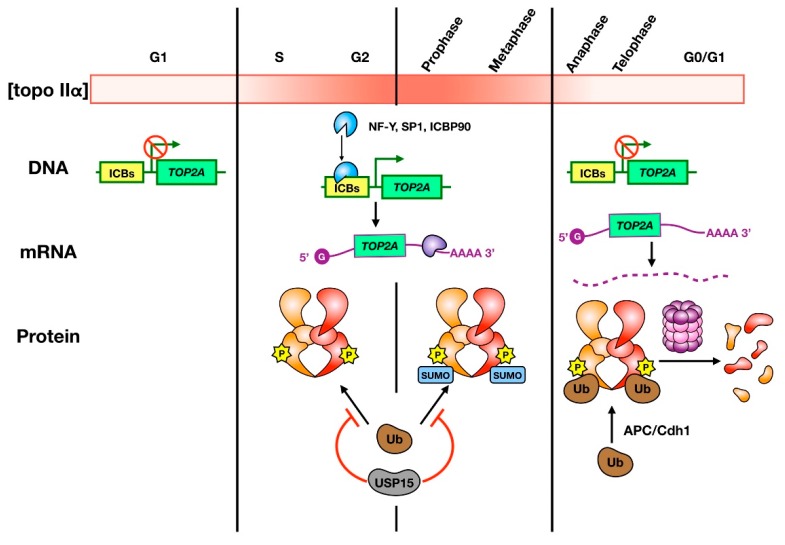
Regulation of topo IIα abundance during the cell cycle. The color gradient bar indicates how levels of topo IIα vary from low (white) to high (red) in different cell cycle stages. The cartoon depicts the current understanding of how topo IIα transcription (DNA), translation (mRNA), and stability (protein) are regulated. Transcription of topo IIα occurs in S phase, through to mitosis, and is regulated by transcription factor binding to ICBs in the promoter region of the *TOP2A* gene. Translation is controlled by cell cycle-dependent mRNA stability, and potentially by cell cycle-dependent proteins that bind and stabilize the 3′-UTR. Topo IIα protein is protected from ubiquitin-mediated degradation during S phase through metaphase by the deubiquitylase USP15. Cell cycle-dependent phosphorylation also occurs from S phase to metaphase and SUMOylation occurs upon entry into mitosis. SUMO modifications are lost upon anaphase onset. At this stage, topo IIα is ubiquitylated by APC/Cdh1 and is subject to proteasomal degradation.

**Figure 2 genes-10-00859-f002:**
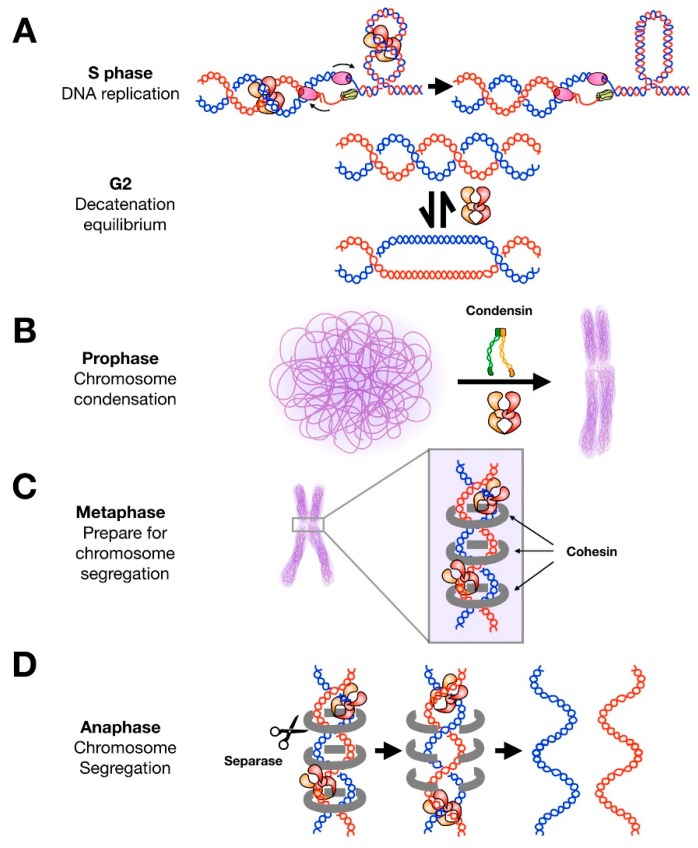
Roles of topo II throughout the cell cycle. (**A**) During DNA replication, topo II decatenates newly-replicated sister chromosomes and aids in relaxing positive supercoils that accumulate ahead of replication forks. Most decatenation is complete upon entry into prophase. Topo II can operate bidirectionally and is still present in high concentrations during G2; it has been suggested that the enzyme may contribute to maintaining a catenated state of sister chromosomes to assist chromosome condensation and cohesion without subsequently interfering with segregation. (**B**) Condensin and topo II are both required for efficient condensation of chromosomes in prophase. (**C**) In metaphase, topo II localizes to centromeres but does not complete decatenation of these regions. It has been suggested that cohesin protects these regions from topo II activity. (**D**) Upon anaphase entry, cohesin is released by separase and topo II rapidly decatenates sister chromatids to allow for chromosome segregation.

**Figure 3 genes-10-00859-f003:**
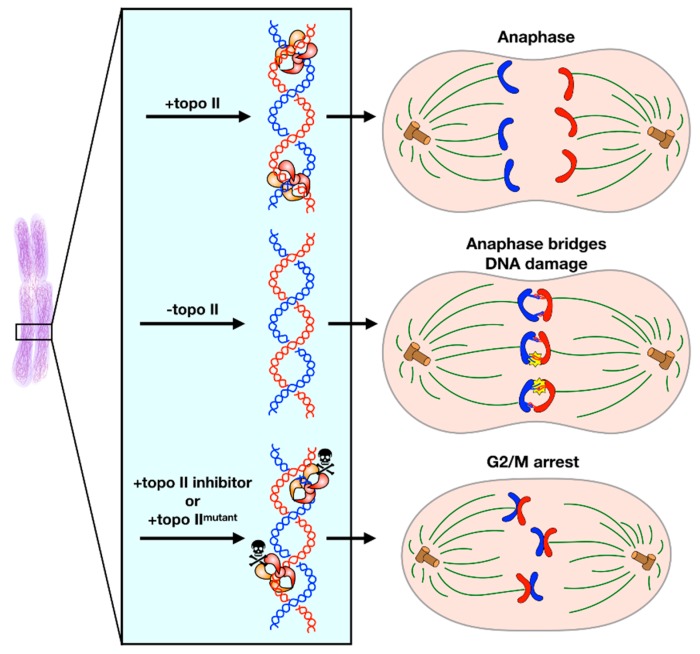
Decatenation Checkpoint. (**Upper**) Under normal conditions, topo II completes decatenation of intertwined sister chromosomes immediately before anaphase to allow for faithful chromosome segregation. (**Middle**) If topo II is absent, the cell cannot sense that sister chromosomes are catenated and will proceed into anaphase. Anaphase entry in this condition leads to the formation of anaphase bridges and DNA damage. (**Lower**) If topo II activity is inhibited by a small molecule or a mutation that impedes enzymatic activity, the cell will arrest in G2/M phase until either decatenation is complete or the cell enters apoptosis.

**Figure 4 genes-10-00859-f004:**
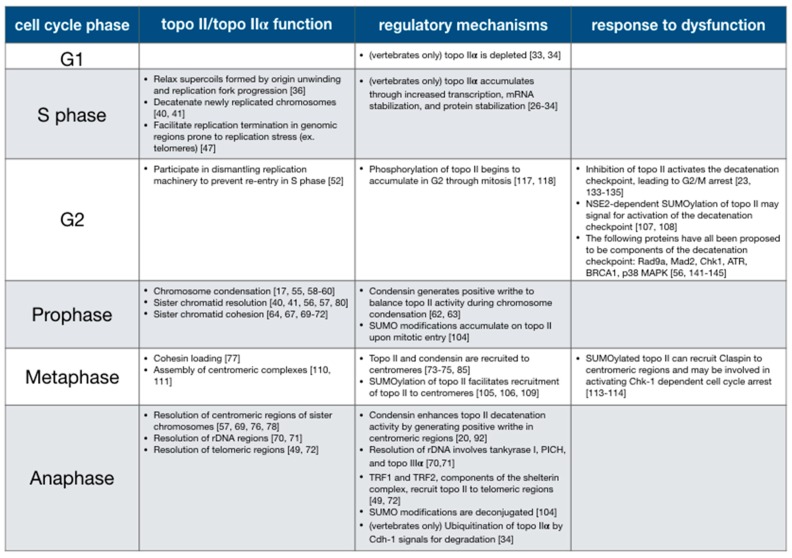
Summary of topo II functions and regulatory mechanisms throughout the cell cycle.

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
