# Peer review of "Cell Cycle-Dependent Control and Roles of DNA Topoisomerase II"

_genes, 2019, doi:10.3390/genes10110859_

Round 1

Reviewer 1 Report

The review is well constructed with a good balance of information, appropriate figure in the related field of interest. However, the manuscript requires a minor revision before it could be considered for publication.

Comments to the authors:

Full name of some abbreviation is missing. In general, symbols for genes are italicized (e.g., IGF1), whereas symbols for proteins are not italicized (e.g., IGF1). The formatting of symbols for RNA and complementary DNA (cDNA) usually follows the same conventions as those for gene symbols. The subsection of Figure 3 is mentioned as upper (section 5), middle (section 5), 3C (section 7), please make it uniform. Either 3A,3B,3C or upper, middle, lower (as in figure legend). A table summarizing various roles of topo II throughout the cell cycle, as well as mechanisms that have been found to govern and/or respond to topo II function and dysfunction would be ideal (with appropriate reference). Can you please check the plagiarism % and make sure it is within the journal limit for plagiarism?

Reviewer 2 Report

Lee and Berger review cell cycle roles of DNA topoisomerase II. The review is well-written and brings together various important concepts that are frequently discussed too briefly in comprehensive reviews of DNA topoisomerases. As frequently happens with this type of review, there are a few points that deserve a more careful description.

Minor points

The statement in lines 68-70 requires references. Although I haven’t found specific references, my sense was that in lower eukaryotes (having only one Top2 isoform) that Top2 expression was proliferation regulated, and non-dividing cells have low Top2 levels. The statement on lines 68-70 is slightly misleading because it is unclear whether a type II topoisomerase is essential for any process besides chromosome segregation at mitosis.

There is no discussion of replication initiation. There is one relevant paper that may merit discussion: Genes Cells. 2004 Oct;9(10):905-17. DNA topoisomerase II is required for the G0-to-S phase transition in Drosophila Schneider cells, but not in yeast.

Line 294 – “Drosophila” should be capitalized.

Line 300 – ICRF183 should be ICRF193?

Line 345 - “SAC activation appears to be unaltered in topo II temperature-sensitive mutants 345 treated with nocodazole, demonstrating that the SAC is independent of topo II activity” I’m not sure I understand the point – The quoted result says Top2 activity is not required to activate the SAC. But Top2 activity is not required to activate a decatenation checkpoint either.

On the question of the decatenation checkpoint, it is important to note that most studies of the decatenation checkpoint rely on ICRF-193. Since ICRF-193 induces DNA damage, at least in some contexts, this discussion needs a bit of qualification.

Two other points regarding the decatenation checkpoint deserve mention: Putative small molecules have been identified that affect this checkpoint Chem Biol. 2003 Dec;10(12):1267-79. Small molecule modulation of the human chromatid decatenation checkpoint. Haggarty SJ1, Koeller KM, Kau TR, Silver PA, Roberge M, Schreiber SL.

The decatenation checkpoint appears absent in stem cells and some progenitor cells. Cancer Cell. 2005 Dec;8(6):479-84. Decatenation checkpoint deficiency in stem and progenitor cells. Damelin M1, Sun YE, Sodja VB, Bestor TH.

Reviewer 3 Report

I almost never do this, but I recommend publishing this paper as is. It is a very useful review paper that the community will appreciate.

Author Response

Reviewer 3 did not suggest any revisions. 

Reviewer 4 Report

In this article, Lee and Berger review the different functions of DNA topoisomerase II mainly in the context of events during the cell-cycle such as DNA replication and mitosis. This is a very well written review from a group that has contributed extensively to the structural and functional understanding of topoisomerases. The only suggestion I have that may improve the manuscript is to include a discussion on the role of topoisomerase II in transcriptional control. This has recently come to the spotlight after the finding that Topo IIβ controls transcription of immediate-early genes during neuronal activity. In the present form, the transcriptional regulatory role of Topo II is only briefly mentioned in one sentence.
